# Maternal body mass index and the risk of early-onset Group B Streptococcus disease in newborns: A systematic review and meta-analysis

Elisabeth Emanuel Graae[1]*, Niels Uldbjerg[2,3], Flemming Skjøth[4,5], Marie Aunskjær Bech[1], Pernille Nathalie Nielsen[1], Susani Rothmann Karkov[1], Caroline Margaret Moos[6], Mohammed Rohi Khalil[1]

1 Department of Gynecology and Obstetrics, Lillebælt University Hospital, Kolding, Denmark, 2 Department of Obstetrics and Gynecology, Aarhus University Hospital, Skejby, Denmark, 3 Department of Clinical Medicine, Aarhus University, Aarhus, Denmark, 4 Department of Regional Health Research, University of Southern Denmark, Odense, Denmark, 5 Research Support Unit, Lillebælt University Hospital, Vejle, Denmark, 6 Department of Clinical Research, Sønderjylland University Hospital, Aabenraa, Denmark

* eligraae@gmail.com

## Abstract

### Background

Early-onset group B Streptococcus disease (EOGBS) remains a leading cause of neonatal morbidity and mortality. The incidence can be substantially reduced by intrapartum antibiotic prophylaxis in women with defined risk factors. However, the role of high prepregnancy body mass index (BMI) as a risk factor remains unclear. This systematic review and meta-analysis therefore aimed to evaluate the association between maternal BMI and the risk of EOGBS, as well as related proxy outcomes, including intrapartum vaginal GBS colonization and rectovaginal or urinary GBS colonization before term.

### Methods

We systematically searched MEDLINE, Embase, and Cochrane CENTRAL on the 28th of January 2026, for studies examining the relationship between pregestational BMI and EOGBS or its proxy outcomes. Eligible studies included observational and interventional designs but not case reports and conference abstracts. Risk of bias was assessed using the QUIPS tool. Random-effects meta-regression and sensitivity analyses were performed.

### Results

We identified 19 eligible observational studies reporting data from a total of 3,707,047 women, encompassing 277,887 cases. For the risk of EOGBS and its proxy

**Data availability statement:** All relevant data are within the manuscript and its Supporting information files.

**Funding:** The author(s) received no specific funding for this work.

**Competing interests:** The authors have declared that no competing interests exist.

outcomes, assuming a log-linear association, our meta-regression showed a 2.4% increase in the odds ratio (OR) per unit increase in BMI. This corresponds to an OR of 1.4 (95% CI 1.1–1.6) for a BMI of 35 and 1.7 (95% CI 1.3–2.3) for a BMI of 45, compared to a normal BMI of 22.3. One very large study on 1,971,346 live singleton births with 780 EOGBS cases, found a hazard ratio of 2.4 (95% CI 1.7–3.4) for a BMI of 35.0–39.9 compared to normal BMI (18.5–24.9).

## Conclusions

Although the overall association appears modest, incorporating BMI may improve prevention strategies for EOGBS

.

---

## 1. Background

Early-onset neonatal infection with Group B Streptococcus (EOGBS), defined as infection occurring within the first 6 days of life, typically presenting as sepsis, pneumonia, or meningitis is a feared and potentially lethal condition. It results from vertical transmission of GBS from mother to infant, typically following rupture of the fetal membranes during labor [1]. The risk of EOGBS can be reduced dramatically by intrapartum antibiotic prophylaxis (IAP) administered to women colonized with vaginal Group B Streptococcus (vGBS) [1]. There are two main strategies for identifying candidates for IAP: a universal screening-based approach, where all pregnant women are screened for GBS colonization (typically at 35–37 weeks gestation), and a risk-based approach, which offers antibiotics only to women with predefined risk factors. The risk-based approach, used in several European countries, relies on risk factors including history of EOGBS in a previous infant, GBS bacteriuria during the current pregnancy, preterm labor, and prolonged rupture of membranes exceeding 12 hours.

Even though vGBS colonization in pregnancy has a global prevalence of approximately 18% [2], the incidence of EOGBS is only 0.75 per 1000 live births in populations not offered IAP [3]. In populations where IAP is offered to women with the above-mentioned risk factors, the incidence is reduced to 0.23 per 1,000 livebirths [3]. However, up to 80% of EOGBS cases in these settings occur in neonates born to women without these risk factors [4,5]. Consequently, there is ongoing discussion regarding how to improve current screening strategies.

Emerging evidence suggests that maternal metabolic factors may also influence colonization and transmission risks. In particular, increasing maternal body mass index (BMI) has gained attention as a potential risk modifier [6,7]. Maternal obesity is known to affect immune responses, hormonal regulation, and microbiome composition, which may in turn affect susceptibility to infections. Large epidemiological studies have demonstrated that obesity is associated with a substantially increased risk of severe infections and infection-related hospitalization across multiple pathogens and organ systems, suggesting a general impairment of host defense mechanisms

in individuals with elevated BMI [8]. This association has been highlighted in a large multicohort study including more than 500,000 participants, which found that higher BMI was associated with markedly increased risks of severe infection outcomes worldwide [8]. Such findings support the biological plausibility that maternal obesity could also influence susceptibility to infections relevant in pregnancy and the perinatal period. This includes not only urinary tract infections and wound infections, but also outcomes used as proxies for EOGBS, such as intrapartum vGBS and rectovaginal GBS colonization before term [9–12]. Given the low incidence of EOGBS, these proxy outcomes are often used as the primary outcome in epidemiologic studies assessing EOGBS risk [6,7].

Therefore, this systematic review and meta-analysis aimed to evaluate the associations between BMI and EOGBS, intrapartum vGBS, and rectovaginal GBS colonization before term.

## 2. Methods

### 2.1 Protocol and registration

This study protocol was registered in PROSPERO (CRD42023439201) and conducted in accordance with the PRISMA 2020 guidelines.

### 2.2 Eligibility criteria

We included original studies of any design (excluding case reports, conference abstracts, and unpublished studies) that reported associations between maternal BMI assessed before 10 weeks of gestation and the outcomes of interest: EOGBS, intrapartum vGBS (vaginal Group B Streptococcus), or rectovaginal/urinary GBS colonization before term. There were no date or language restrictions.

The PFO structure [13] was used to determine the suitability of article inclusion into the study:

• Population: Pregnant women.

• Factor: Pregestational BMI assessed before the 10th gestational week.

• Outcome: EOGBS or its proxy outcomes, including intrapartum vGBS (vaginal Group B streptococcus), and rectovaginal or urinary GBS colonization before term.

The prognostic factor was defined as pregestational BMI $\geq 25\,kg/m^2$ compared with a reference BMI defined as the traditional "normal" range BMI of 18.5 to 24.9 $kg/m^2$. Studies reporting BMI in multiple categories were included to allow assessment of a potential dose–response relationship between increasing BMI and GBS-related outcomes.

### 2.3 Search strategy

A comprehensive search was conducted in MEDLINE (Ovid), Embase (Ovid), and CENTRAL (Cochrane Library) from database inception to the 28th of January 2026. Grey literature was identified through Google Scholar, and citation tracking was conducted using Web of Science. The Google Scholar search included the terms "BMI", "risk", "GBS", and "EOGBS", and screening was conducted through the first 30 pages (300 results). Moreover, references cited by the included studies were also screened for eligibility. The search strategy combined the terms: pregnancy, BMI/obesity, and GBS-related outcomes. Our search strategy included controlled vocabulary Medical Subject Headings (MeSH) and free-text terms, combined in three blocks using the Boolean operator "AND". In each block, exploded medical subject headings or equivalents, depending on the database, truncations, free-text words, and narrower terms operators were used as appropriate. The following MeSH terms were included in Embase and MEDLINE; *high risk pregnancy, pregnancy, pregnancy complications, childbirth, obesity, morbid obesity, body mass, body mass index, body mass, maternal obesity, streptococcal infections* and *streptococcus agalactiae.* The complete search strategy is available in S1 Table.

## 2.4 Study selection and data extraction

Two reviewers, PN/MB and EG, independently screened titles, abstracts, and full texts using Covidence [14]. Duplicates were removed using Covidence's automated deduplication algorithm. Eligible studies were selected based on predefined PFO (Population, Factor, Outcome) criteria [13]. Two reviewers, MB and EG, independently extracted data in a pre-defined, partly adjusted form on study design, setting, BMI categorization, GBS detection methods, population demographics, and outcomes. Inconsistencies in both study selection and data extraction were settled by consensus. There was no need for a third-party adjudication.

Data extraction was based on information reported in the published articles and their supplementary materials. When highly relevant details were missing, attempts were made to contact the corresponding authors to obtain additional information. However, no responses were received. Therefore, the analysis relied on the data available in the published sources.

## 2.5 Statistical analysis

The association between BMI and the risk of GBS across studies was analyzed using a random-effects meta-regression [15] of the log odds ratio with standard errors estimated from the reported 95% confidence intervals. As the reported BMI intervals varied substantially between studies, we translated the intervals to a continuous scale by estimating the within-interval mean BMI through numerical integration, assuming a universal log-normal distribution [16] with a mean of log (24.6) and a standard error of log (1.2) based on data from the Danish Medical Birth Registry [17,18]. The reference categories representing normal weight were included with a low fixed standard error (0.001 on a log scale). A sensitivity analysis was performed by assuming alternative BMI distributions (normal and uniform) to test the robustness of the log-normal assumption used for interval estimation. Reported adjusted odds ratios were used in the main analysis when available to account for potential confounders. Since there is not yet a strong consensus on the key confounders of the association in question, we did not use a pre-specified approach. Cohort studies reporting hazard ratios were included, assuming that hazard ratios approximate the odds ratio, given the rare incidence of EOGBS. Due to overlapping cohorts in the two Swedish studies [6,7] evaluating BMI as an independent risk factor for EOGBS, the study with the shortest inclusion period, by Håkansson et al. [6] was excluded from the meta-analysis. Subgroup analyses for each proxy outcome were performed, as well as sensitivity analyses restricting results to studies reporting adjusted ORs using only crude OR, and leave-one-out analyses [19]. Further subgroup analyses were conducted according to the GBS testing methodology, including anatomical sampling site (rectovaginal vs mixed/other) and study design. The estimated OR with 95% confidence intervals for the relative risk of outcome per one unit change in BMI was reported, as well as the association between BMI and OR of outcome was plotted with the average normal BMI as reference (BMI = 22.3). Heterogeneity was assessed using Cochrane's I² statistic [20]. A naive categorization of values for I² would not be appropriate for all circumstances, although we would tentatively assign adjectives of low, moderate, and high to I² values of 25%, 50%, and 75%. Publication bias was evaluated using residual funnel plots. The procedure meta meregress and own programs in *Stata Statistical Software: Release 18.5* and College Station, TX: StataCorp LLC were used for the analyses.

## 2.6 Quality assessment

The quality of the studies was assessed using the Quality of Prognosis Studies in Systematic Reviews (QUIPS) tool [21]. The QUIPS tool is developed for studies on prognostic factors and supports a systematic appraisal of bias across six key domains: study participation, attrition, prognostic factor measurement, outcome measurement, confounding, and statistical analysis. Domain-specific decision rules were predefined prior to the assessment. In each domain, 3–6 subdomains were considered in accordance with guides for prognostic research [21]. Detailed reporting of prognostic factor measurement varied across studies due to their observational design. Given that BMI is a routinely used and standardized measure in

epidemiological research, the lack of reporting of the assessment method was judged as moderate rather than high RoB (Risk of Bias).

Particular emphasis was placed on confounding and statistical analysis, as these were considered critical to prognostic research. Adequate statistical adjustment for confounding was required, including multivariable modelling with clearly reported effect estimates. Yet risk factors for GBS and relevant confounders remain incompletely established; however, based on previous research, GDM, diabetes, parity, socioeconomic status, and ethnicity were considered important potential confounders and integrated into the pragmatic RoB evaluation of the confounding domain. Studies reporting only univariable analyses were rated as high risk in the confounding domain.

Studies were classified as:

- **Low overall RoB:** low risk in all key domains (including confounding and statistical analysis/reporting) and no more than one domain rated as moderate risk.

- **Moderate overall RoB**: moderate risk in one or more key domains without any domain rated as high risk.

- **High overall RoB**: high risk in at least one key domain or multiple domains rated as moderate risk.

The assessment was performed by two authors, SK and EG, working independently, followed by evaluation. Disagreements were resolved through discussion until a consensus was reached.

## 3. Results

### 3.1 Main results

The search strategy identified 826 records after duplicate records were removed (Fig 1). The grey literature and citation tracking did not yield any additional studies. Of the 826 records, 807 were excluded for the following reasons: reporting only the mean BMI; comparison of a BMI > 23 (within the normal range) with BMI < 23; BMI categories not meeting the eligibility criteria (BMI < 30 compared to BMI > 35); BMI not reported to GBS status; study population with BMI < 25, EOS (Early Onset neonatal Sepsis) outcomes not differentiating the infectious agent; or inappropriate study design, most commonly unpublished studies.

The 19 eligible studies reported data from a total of 3,707,047 women, encompassing 277,887 cases of either EOGBS (two studies [6,7]) or proxy outcomes (17 studies) (see Fig 1 and Table 1). Due to overlap between the EOGBS cohorts, only one study [7] was included in the primary analysis. The studies reporting proxy EOGBS outcomes were categorized according to the timing of GBS assessment: intrapartum (one study (5.3%) [22]); at gestational weeks 35–37, as in universal routine screening (e.g., in the US) (seven studies (36.8%) [12,23–28]); and at other time points during pregnancy (nine studies, (47.4%) [9–11,29–34]). Among studies assessing GBS in gestational weeks 35–37, routine universal screening was implemented in two of seven study settings [12,26]. The studies included nine European, five North American, three Asian, and two African populations, and all had either a retrospective or cross-sectional design. Study characteristics are presented in Tables 1 and 2.

In the two Swedish nationwide cohort studies reporting EOGBS outcomes [6,7] the diagnosis was mainly based on culture-confirmed early-onset sepsis. The proxy outcomes used in the remaining studies were based on either intrapartum vaginal GBS-PCR; one study (5.3%) [22], rectovaginal culture; 10 studies (52.6%) [23–30,34,32], vaginal culture; one study (5.3%) [33] or mixed urinary/rectovaginal/vaginal GBS positive samples; five studies (26.3%) [9–12,31]. Common covariates considered potential confounders or effect modifiers such as maternal age, parity, smoking status, diabetes, race, socioeconomic status, hypertension, and preeclampsia (Table 3). BMI categorization varied considerably across studies, as illustrated in Table 3, which presents the studies' main findings. When available, we based our analyses on adjusted rather than crude outcome measures, as emphasized in Table 3.

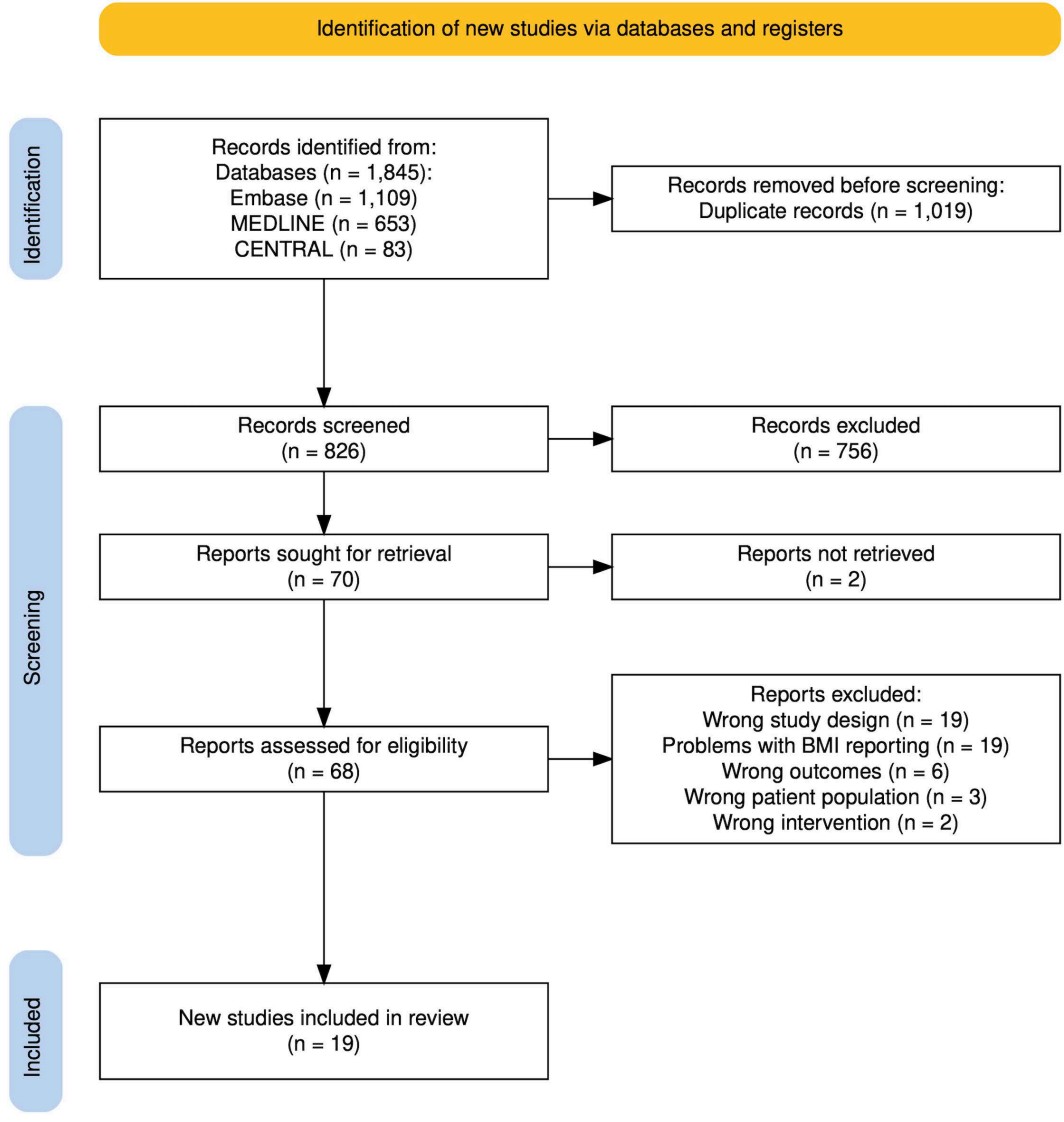

**Fig 1. PRISMA flow-chart on the selection process.**

Our primary analysis, presented in Fig 2, shows the OR of EOGBS or proxy outcomes as a function of BMI with normal BMI at 22.3.

In the primary meta-analysis 18 studies were included. Assuming a log-linear association, each one-unit increase in BMI was associated with 2.4% higher odds of EOGBS and proxy outcomes (OR 1.024, 95% CI 1.010–1.037; p = 0.001). Sensitivity analyses showed consistent findings. The pooled OR based on crude estimates of all 18 studies was 1.024 (95% CI 1.009–1.041; p = 0.002), whereas the pooled OR based on adjusted estimates, including 14 studies, was 1.030 (95% CI 1.017–1.042; p < 0.001) per BMI unit (Fig 3).

When translated to clinically relevant BMI differences, these estimates correspond to approximately 20% higher odds at a BMI of 30 compared with 22.3, and about 52% higher odds at a BMI of 40. Estimated ORs across selected BMI levels are presented in Table 4.

**Table 1. Characteristics of included studies, geographical and temporal setup, size, design, and detection methods.**

| Author, Year | Country | Study Period | Sample size (Cases) | Study Design | Inclusion Criteria | GBS Detection Method |
|---|---|---|---|---|---|---|
| Outcome: EOGBS | | | | | | |
| Villamor, 2021 [7] | Sweden | 1997-2016 | 1,971,346 (866) | Nationwide, multi-center cohort, | GA > 22 weeks, admission within 72 hrs | Blood culture-confirmed early-onset sepsis |
| Håkansson, 2008 [6]* | Sweden | 1997-2001 | 344,127 (248) | Nationwide, multi-center cohort, | GA ≥ 23 weeks, vaginal or emergency caesarean | Clinical diagnosis or blood culture |
| Outcome: Intrapartum GBS colonization | | | | | | |
| Khalil, 2022 [22] | Denmark | 2013-2014 | 902 (155) | Single-center, cohort | GA > 37 weeks, age > 18, no antibiotics post-35 weeks | Rectovaginal culture PCR |
| Outcome: Antepartum GBS colonization week 35–37 (universal screening) | | | | | | |
| Alvareza, 2017 [12] | USA | 2013 | 2,045 (594) | Single-center, cohort | GA > 37 weeks, age 18–45, singleton pregnancy | Rectovaginal/ Urine culture |
| Rao, 2019 [23] | UK | 2014-2015 | 6,309 (1,836) | Single-center, cross-sectional | GA 35–37 weeks | Self-collected Rectovaginal culture |
| Rick, 2017 [24] | Guatemala | 2015 | 896 (155) | Single-center, cross-sectional | GA ≥ 35 weeks, age 15–45 | Rectovaginal culture |
| Namugongo, 2016 [25] | Uganda | 2015 | 309 (89) | Single-center, cross-sectional | GA ≥ 35 weeks, attended antenatal clinic | Anovaginal swabs, rapid GBS test kit |
| Bognar, 2025 [26] | Belgium | 2012-2021 | 1,081,085 (199,669) | Nationwide, multi-center, retrospective cohort | Live births | Rectovaginal culture |
| Zhou, 2024 [27] | China | 2019-2021 | 3269 (231) | Single-center, cohort | GA 35–37, no vaginal infections or prior use of antimicrobial drugs, no use of vaginal washes or suppositories within 24 hours prior to sampling, no recent sexual activity or use of antimicrobial drugs within the past 2 weeks before screening, no coexistence of malignant disease or impaired liver or kidney function | Rectovaginal culture |
| Nguyen, 2025 [28] | Vietnam | 2021-2022 | 876 (178) | Multi-center, cross-sectional | GA 35–37 | Rectovaginal culture |
| Outcome: Antepartum GBS colonization any time in pregnancy | | | | | | |
| Manzanares, 2012 [29] | Spain | 2007-2009 | 3,016 (399) | Single-center, cohort | Singleton pregnancy | Rectovaginal culture |
| Manzanares, 2019 [11] | Spain | 2012-2014 | 9,877 (1,835) | Single-center, cohort | GA > 26 weeks, singleton pregnancy, live births | Rectovaginal swab, GBS bacteriuria |
| Kwon, 2024 [30] | USA | 2010-2017 | 8,019 (997) | Single-center, cross-sectional | GA > 20 weeks, Black race, no chronic hypertension or diabetes | Rectovaginal culture |
| Dahan-Saal, 2011 [31] | France | 2004-2007 | 2,911 (cases) | Single-center, case-control | GA ≥ 24 weeks, singleton pregnancy | Vaginal/Urine culture, newborn sample |
| Kleweis, 2015 [32] | USA | 2004-2008 | 7,711 (1,989) | Single-center, cohort | GA > 37 weeks, singleton pregnancy | Rectovaginal culture |
| Venkatesh, 2020 [9] | USA | 2002-2008 | 115,070 (23,625) | Multi-center, cohort | GA ≥ 37 weeks | Urine, Vaginal, or Rectovaginal cultures |
| Stapleton, 2005 [10] | USA | 1997-2002 | 40,459 (cases) | Multi-center, case-control | Singleton pregnancy | GBS colonization from discharge record |

*(Continued)*

**Table 1.** (Continued)

| Author, Year | Country | Study Period | Sample size (Cases) | Study Design | Inclusion Criteria | GBS Detection Method |
|---|---|---|---|---|---|---|
| Chen, 2018 [33] | China | 2015 | 2,121 (104) | Multi-center, cross-sectional | GA ≥ 28 weeks | Vaginal culture |
| Melchor, 2019 [34] | Spain | 2013–2017 | 16,609 (1,547) | Historical Single-center, cohort | GA ≥ 23 weeks, singleton pregnancy | Rectovaginal culture |

*This study was not included in the primary analysis due to overlapping cohorts with Villamor et al, 2021 [7].

GA: Gestational age.

GBS: Group B streptococcus.

EOGBS: Early-onset group B streptococcus disease.

In S2 Table, absolute risk differences corresponding to the estimated OR for the BMI levels reported in Table 4 are given considering a background level of outcome of 0.43 per 1000 [3] for normal BMI. A BMI level of 40 represents a risk difference of 0.22 (95% CI: 0.07–0.37) in EOGBS or proxy outcomes per 1000.

Two studies, both included in the primary analysis, demonstrated markedly stronger associations (Fig 2). One of these [7], assessed as having a low risk of bias, was based on a large Swedish cohort and used EOGBS as the outcome. It reported an adjusted hazard ratio of 2.4 (95% CI 1.7–3.4) for BMI 35–39.9, with an even stronger association observed among term infants. In absolute terms, the unadjusted equivalent corresponds to a risk of 0.86 per 1000 live births among individuals with BMI 35–39.9 compared with 0.39 per 1000 among those with normal BMI (18.5–24.9), corresponding to an absolute risk difference of approximately 0.47 per 1000 births. The other study [22], assessed as having a moderate risk of bias, examined intrapartum GBS colonization and reported an OR of 2.68 (95% CI 1.26–2.72) for BMI ≥ 30. Overall, findings were broadly consistent but not uniform. Kwon et al. reported an inverse association [30], whereas Rick et al. and Rao et al. reported non-significant or attenuated associations [23,24]. Similarly, Zhou et al. and Nguyen et al. reported non-significant results, although this may reflect the lower BMI threshold used to define exposure (BMI > 25) [27,28]. Despite these differences, leave-one-out analyses indicated that no single study had a decisive influence on the main analysis (S1 Fig).

The influence of sampling method, study design, and model assumptions was examined via a range of sensitivity analyses (Fig 3), which showed minor changes in the pooled ORs. In particular, altering the assumed underlying distribution of BMI led to some changes; however, both distributions deviate markedly from the known skewed population distribution among pregnant women.

The heterogeneity between the studies was substantial with an $I^2$ of 75.4% (Fig 3). However, in studies analyzing rectovaginal cultures at gestational weeks 35–37, $I^2$ was only 3.5% [12,23–28]. For the proxy outcomes – based on seven studies assessing GBS colonization at or after the 35th gestational week [12,23–28] and nine studies assessing GBS status at any time during pregnancy [9–11,29–34] – only studies with limited weight in the primary analysis showed results that deviated significantly from the overall findings (Fig 2).

### 3.2 Risk of bias assessment

The risk of bias (RoB) was evaluated and is available in Table 5.

According to the QUIPS assessment, seven studies were rated as having an overall low RoB, eight studies as moderate, and four as high [12,27,30,32]. Primarily, RoB issues were related to study attrition, selection bias, risk of confounding, or missing information on exposure and outcome characteristics (Table 5). Reporting of BMI assessment methods was generally limited. Pregestational BMI was most likely frequently based on self-report. Information on GBS

**Table 2. Characteristics of the study population.**

| Author, Year | Age (mean ± SD) | GA | GDM (%) | Parity (P0%)** | Ethnicity | Socio-economic status | Smoking (%) |
|---|---|---|---|---|---|---|---|
| Outcome: EOGBS. | | | | | | | |
| Villamor, 2021 [7] | 30.1 (5.1) | 37-41: 88% | 1.6% | 58% | Nordic 80%, Non-nordic 19.9% | High: 37.9% ≥15 years education | 8.7 |
| Håkansson, 2008 [6]*** | NA | NA | NA | NA | Swedish population | NA | NA |
| Outcome: Intrapartum GBS colonization | | | | | | | |
| Khalil, 2022 [22] | 87.1% age 25–40 | 37-40: 42.6%, ≥40: 57.4% | NA | 37.2% | Danish population | NA | 9.7 |
| Outcome: Antepartum GBS colonization week 35–37 (universal screening) | | | | | | | |
| Alvareza, 2017 [12] | 27.9±5.1 | 39.3±1.3 | 10.8%* | NA | White 26.5%, African American 48.7%, Hispanic 20.7%, Asian 2.6% | Marital status: Single 58% Married 39.5% Divorced 2.5% | 12.4 |
| Rao, 2019 [23] | 30.6±5.6 | NA | NA | 46% | White British 8.9%, White other 27.6%, Indian subcontinent 33.9%, Asian other 13.1%, Black 8.9%, Other 7.6% | IMD Quintiles: Q1 8.5% Q2 29.6% Q3 32.4% Q4 21.5% Q5 7.9% | NA |
| Rick, 2017 [24] | 25.4±6.6 | ≥35: 100% | 1.6%* | 32.5% | Non-indigenous 90.2%, Native Mayan 9.8% | Education: Primary 62.1%, Secondary 27.5% Marital status: Married 30.5%, single 16.0%, partnership 53.5% Income ≤200 $/month 94.3% Non-literate 6.0% | NA |
| Namugongo, 2016 [25] | 25 | 35–37.6: 54.4% 38-40.6: 36.9% ≥41: 8.7% | NA | 37.2% | Banyankore 75.4%, Bakiga 9.7%, Baganda 12.3% | Education: None 8.0% Primary 34.3%, Secondary 38.8%, Tertiary 24.3% Marital status: Married 92.6%, not married 7.4% No employment: 23.0% | NA |
| Bognar, 2025 [26] | 30.6 | 38.8 | NA | 43.6% | Belgium 77.2%, Northern/Southern/Western Europe 7.3%, Eastern Europe/Russia 5.4%, North Africa 3.8%, Sub-Saharan Africa 3.1%, Other 3.3% | Education: Approximately 75% upper secondary or higher Work status: Paid employment or student 69.8% | NA |
| Zhou, 2024 [27] | <35: 94.9% ≥35: 5.1% | 35-37 | NA | NA | Chinese population | Below college degree: 59.3% | NA |
| Nguyen, 2025 [28] | 29.3±4.8 | 35-37 | NA | NA | Kinh ethnic group 97.3% | Education: Above vocational training or a bachelor's degree: 82.9% | NA |
| BMI and risk of antepartum GBS colonization any time in pregnancy | | | | | | | |
| Manzanares, 2012 [29] | 29.4±5.2 | ≥37: 87.4% <33: 2.6% | 10% | 43.1% | NA | NA | NA |
| Manzanares, 2019 [11] | 30.65±5.7 | 276.7±17.1 days | 5.9% | 34.7% | Caucasian 98.4%, Black 0.8%, Asian 0.6% | NA | 14.8 |

*(Continued)*

**Table 2.** (Continued)

| Author, Year | Age (mean±SD) | GA | GDM (%) | Parity (P0%)** | Ethnicity | Socio-economic status | Smoking (%) |
|---|---|---|---|---|---|---|---|
| Kwon, 2023 [30] | <20: 7.8%, 20-29: 52.1%, 30-40: 35.9%,>40: 4.2% | ≥37: 77.5% | 0% | 37.3% | Black 100% | NA | 3.3 |
| Dahan-Saal, 2011 [31] | 27.8±6 | ≥37: 91% | 8.3% | 39.8% | NA | Low: 69.3% ≤ high school, 45.2% unemployed Marital status: Married 32.4%, single 65.6%, divorced/widow 1.6% | NA |
| Kleweis, 2015 [32] | 25.0±6.1 | 39.0±1.2 | NA | 37.6% | Black 71.5%, Caucasian 19.6% | NA | 18 |
| Venkatesh, 2020 [9] | 27.2±6.0 | 38.9±1.1 | 4.2% | 6.3% | White 54.8%, Black 19.7%, Latina 19.3% | Insurance: Private 63.2%, public 30.1% Not married: 35.7% | 6.3 |
| Stapleton, 2005 [10] | 27.5±6.2 | NA | 3.8% | 44.4% | White 71.1%, Black 4.2%, Native American 2.5%, Hispanic 13.5% | Education: High school graduate or more 52.2%, any high school 43.4%, no high school 4.4% Annual income of mother's census tract of residence~44,000 dollars | 11.5 |
| Chen, 2018 [33] | 27.7±4.7 | 39.4±1.4 | NA | NA | NA | Education: Junior high 38.5%, Senior high 31.9% Family monthly income: < 435 $ 12.8%, ≥1159$ 17.4% | NA |
| Melchor, 2019 [34] | 33.9±4.9 | 39.0±1.9 | 5.3% | 57% | NA | NA | 13.1 |

* Pregestational and gestational diabetes pooled.

** Percentage of nulliparous participants. P0 = nulliparous.

*** This study was not included in the primary analysis due to overlapping cohorts with Villamor et al, 2021 [7].

GA: Gestational age in weeks.

GDM: Gestational diabetes mellitus.

PE: Preeclampsia.

NA: Not available.

BMI: Body mass index.

HSV: Herpes simplex virus.

GBS: Group B streptococcus.

EOGBS: Early-onset group B streptococcus disease.

IMD: Index of Multiple Deprivation, a metric used to measure area-level socio-economic deprivation based on income, employment, health, education, and other factors. Quintile 1 (Q1) refers to the most deprived (top 20%), quintile 5 (Q5) to the least deprived (bottom 20%).

assessment procedures—including sampling method, anatomical site, handling of multiple sampling sites, and microbiological analysis—was insufficiently described in several studies.

The regression funnel plot in Fig. 4 did not reveal substantial asymmetry, indicating neither publication bias nor between-study heterogeneity.

**Table 3. Study-reported associations between BMI categories (intervals) and outcomes, with a summary of the study conclusions.**

| Study (Year) | Purpose | BMI Categories (%) | Effect size | Statistical method | Adjusted covariates | Summary Conclusion |
|---|---|---|---|---|---|---|
| Outcome: EOGBS | | | | | | |
| Villamor, 2021 [7] | BMI & risk of early-onset sepsis (EOS) | <18.5: 2.2% 18.5-24.9: 64.6% 25-29.9: 22.6% 30-34.9: 7.5% 35-39.9: 2.3% ≥40: 0.8% | 0.73 (0.41-1.31) Ref 1.2 (1.01-1.43) 1.7 (1.35-2.16) 2.4 (1.70-3.41) 2.6 (1.49–4.38) | Adjusted Hazard Ratio (95% CI)for EOGBS: | Maternal age, country of origin, education level, cohabitation with partner, parity, height, smoking, year of delivery | Higher BMI increases risk of EOS. Strong dose-response, particularly for term infants. |
| Håkansson, 2008 [6]* | BMI & neonatal EOGBS | <18.5: 2.6% 18.5-24.9: 63.9% 25-29.9: 24.2% ≥ 30: 9.3% | 1.5 (0.6-3.8) Ref 1.3 (0.9-2.0) 1.8 (1.1-3.0) | Adjusted Odds Ratio (95% CI) | Maternal age, parity, GA, GDM | BMI ≥ 30 significantly increases risk of EOGBS. Effect is weaker but remains significant when including clinical diagnoses. |
| Outcome: Intrapartum GBS colonization | | | | | | |
| Khalil, 2022 [22] | BMI & intrapartum GBS colonization | <18.5: 4.0% 18.5-24.9: 59.0% 25-29.9: 24.0% ≥ 30: 13.0% | 0.26 (0.06-1.11) Ref 1.62 (1.10-2.38) 2.68 (1.26-2.72) | Adjusted Odds Ratio (95% CI) | Parity, GA, vaginal delivery, smoking | Overweight and obesity are associated with increased GBS colonization, impacting the risk-based prevention strategy. |
| Outcome: Antepartum GBS colonization week 35–37 (universal screening) | | | | | | |
| Alvareza, 2017 [12] | BMI & GBS at 35–37 weeks | <25 25–29.9: 25.7% 30–34.9: 29.4% 35–39.9: 38.0% ≥40: 38.6% | Ref 0.97 (0.74–1.26) 1.14 (0.86–1.52) 1.43 (1.01–2.03) 1.27 (0.90–1.79) | Adjusted Odds Ratio (95% CI) | Race, marital status, insurance, chronic hypertension, maternal genital infections | Severe obesity appears associated with higher GBS risk, but the link weakens after adjusting for other variables. |
| Rao, 2019 [23] | BMI & GBS across racial groups | <18.5: 23.0% 18.5-24.9: 64.6% 25–29.9: 29.2% 30–34.9: 31.2% ≥35: 35.2% | 0.76 (0.53-1.07) Ref 1.04 (0.92-1.18) 1.15 (0.97-1.36) 1.38 (1.09-1.74) | Odds Ratio (95% CI) | % | Highest GBS colonization rates found in Black women and those with higher BMI. |
| Rick, 2017 [24] | BMI & GBS across racial groups | <30: 69.6% ≥30: 30.4% | Ref 0.78 (0.52–1.18) | Adjusted Odds Ratio (95% CI) | Maternal age, parity, education, marital status, literacy, ethnicity, home location, smoking, BMI, diabetes, prenatal care utilization, number of sexual partners, prematurity, previous infant with low birth weight or poor infant outcome | No significant association with BMI; GBS prevalence was high overall. |
| Namu-gongo, 2016 [25] | BMI & anogenital GBS | <25: 35.0% 25-29.9: 50.0% ≥30: 15.0% | Ref 1.05 (0.58-1.9) 3.78 (1.78–8.35) | Adjusted Odds Ratio (95% CI) | Maternal age, BMI, residence type, previous perineal tear | Obesity was the only significant predictor of anogenital GBS colonization. |
| Bognar, 2025 [26] | Risk factors of GBS colonization and neonatal infection | ≤18.5: 4.8% 18.5-25: 58.6% 25-30: 23.2% >30: 13.2% | 0.93 (0.91-0.95) Ref 1.05 (1.04-1.06) 1.11 (1.10-1.14) | Adjusted Risk Ratio converted to adjusted Odds Ratio (95% CI) | Maternal age, parity, hypertension, diabetes, maternal education level, maternal work status, maternal nationality | Following risk factors for GBS colonization were confirmed; African origin, parity, obesity and low educational level |
| Zhou [27] | Risk factors, genotypic diversity, and antibiotic resistance patterns of GBS | <25 >25 | Ref 1.02 (0.73-1.44) | Odds Ratio (95% CI) | % | Key risk factors of GBS colonization were identified; absence of pregnancy vomiting, having a college degree or above and biochemic factors, e.g., elevated albumin and bilirubin. |

*(Continued)*

**Table 3.** (Continued)

| Study (Year) | Purpose | BMI Categories (%) | Effect size | Statistical method | Adjusted covariates | Summary Conclusion |
|---|---|---|---|---|---|---|
| Nguyen [28] | Prevalence, risk factors and sero-type distribution of GBS | <25<br>>25 | Ref<br>1.09 (0.78-1.52) | Odds Ratio (95% CI) | % | Higher educational attainment and the presence of yellow vaginal discharge were associated with increased risk of GBS colonization. |
| **Outcome: Antepartum GBS colonization any time in pregnancy** | | | | | | |
| Manzanares, 2012 [29] | BMI & perinatal outcomes | <18.5: 5.5%<br>18.5-34.9: 86.2%<br>>35: 8.3% | 0.72 (0.38-1.39)<br>Ref<br>1.57 (1.15–2.13) | Adjusted Odds Ratio (95% CI) | Maternal age, parity, hypertension, diabetes | Obesity increased GBS and perinatal risks; underweight linked to low birth weight and oligohydramnios. |
| Manzanares, 2019 [11] | BMI & GBS colonization | <25: 62.6%<br>25-29.9: 24.6%<br>>30: 12.8% | 1.05 (0.92-1.21)<br>Ref<br>1.33 (1.12–1.56) | Adjusted Odds Ratio (95% CI) | GA, parity, delivery mode, smoking, diabetes (pregestational and gestational) race and fetal birthweight | Obese women were 33% more likely to be colonized with GBS. EOGBS not directly evaluated. |
| Kwon, 2023 [30] | GBS & preeclampsia (Black women) | <25: 14.7%<br>25-30: 30.5%<br>>30: 54.8% | Ref<br>0.73 (0.6-0.89)<br>0.70 (0.58-0.84) | Odds Ratio (95% CI) | % | GBS-positive women had a significantly reduced risk of preeclampsia. |
| Dahan-Saal, 2011 [31] | BMI & vertical transmission | <18.5: 10%<br>18.5-24.9: 60.7%<br>25-29.9: 17.9%<br>≥30: 11.4% | 0.81 (0.69-0.95)<br>Ref<br>1.06 (0.94-1.2)<br>1.19 (1.03-1.38) | Adjusted Odds Ratio (95% CI) | Country of origin, prenatal care commitment | Obesity increases both GBS colonization and transmission risk to newborns. |
| Kleweis, 2015 [32] | BMI dose-response & GBS | <30: 41.8%<br>30-39.9: 43.4%<br>≥40: 14.8% | Ref<br>1.29 (1.15-1.45)<br>1.54 (1.32–1.79) | Adjusted Odds Ratio (95% CI) | Race, parity, diabetes | Higher BMI showed a clear dose-response relationship with increased GBS colonization. |
| Venkatesh, 2020 [9] | BMI & GBS colonization | <25: 61.0%<br>25-29.9: 22.3%<br>30-34.9: 9.8%<br>35-39.9: 4.2%<br>≥40: 2.7% | Ref<br>1.09 (1.05-1.13)<br>1.2 (1.15-1.26)<br>1.42 (1.33-1.51)<br>1.50 (1.38–1.62) | Adjusted Odds Ratio (95% CI) | Maternal age, parity, pregestational diabetes, insurance status, study site, year of delivery | Risk of GBS colonization increases with each BMI category, especially in higher obesity classes. |
| Stapleton, 2005 [10] | Risk factors for GBS | <18.5: 4.3%<br>18.5-24.9: 54.6%<br>25-29.9: 23%<br>30-39.9: 15.2%<br>>40: 2.9% | 0.98 (0.88-1.10)<br>Ref<br>1.07 (1.01-1.12)<br>1.20 (1.13-1.28)<br>1.45 (1.28–1.63) | Adjusted Odds Ratio (95% CI) | Ethnicity, BMI, diabetes, parity, maternal age, alcohol, income, education, occupation smoking, prenatal care use, year of delivery | Higher BMI is modestly but significantly associated with GBS colonization. |
| Chen, 2018 [33] | GBS in late pregnancy | ≤28: 98,7%<br>>28: 1.3% | Ref<br>3.79 (1.28–11.26) | Adjusted Odds Ratio (95% CI) | BMI, GA, induced abortion, lotion use before pregnancy | Strong association between BMI > 28 and GBS in a Chinese cohort; small sample size. |
| Melchor, 2019 [34] | Obesity & perinatal outcomes | <30: 82.6%<br>≥30:18.4% | Ref<br>1.30 (1.14–1.47) | Adjusted Odds Ratio (95% CI) | Maternal age, parity, hypertension, gestational age | Maternal obesity increases risk of maternal and neonatal complications, including GBS. |

*This study was not included in the primary analysis due to overlapping cohorts with Villamor et al, 2021 [7].

BMI: Body mass index.

GBS: Group B streptococcus.

EOGBS: Early-onset group B streptococcus disease.

 

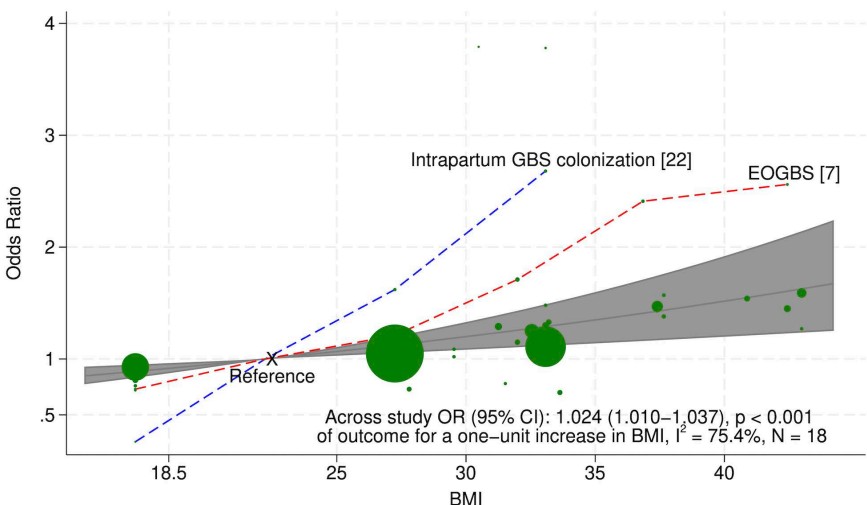

**Fig 2. Association between BMI and EOGBS and proxy outcomes based on random-effects meta-regression.** Depicts ORs for EOGBS or proxy outcomes as a function of BMI, with normal BMI at 22.3 as reference, with the shaded region representing 95% confidence interval based on random-effects meta-regression. Bubbles represent individual studies with diameters inversely proportional to the standard errors of the reported associations.

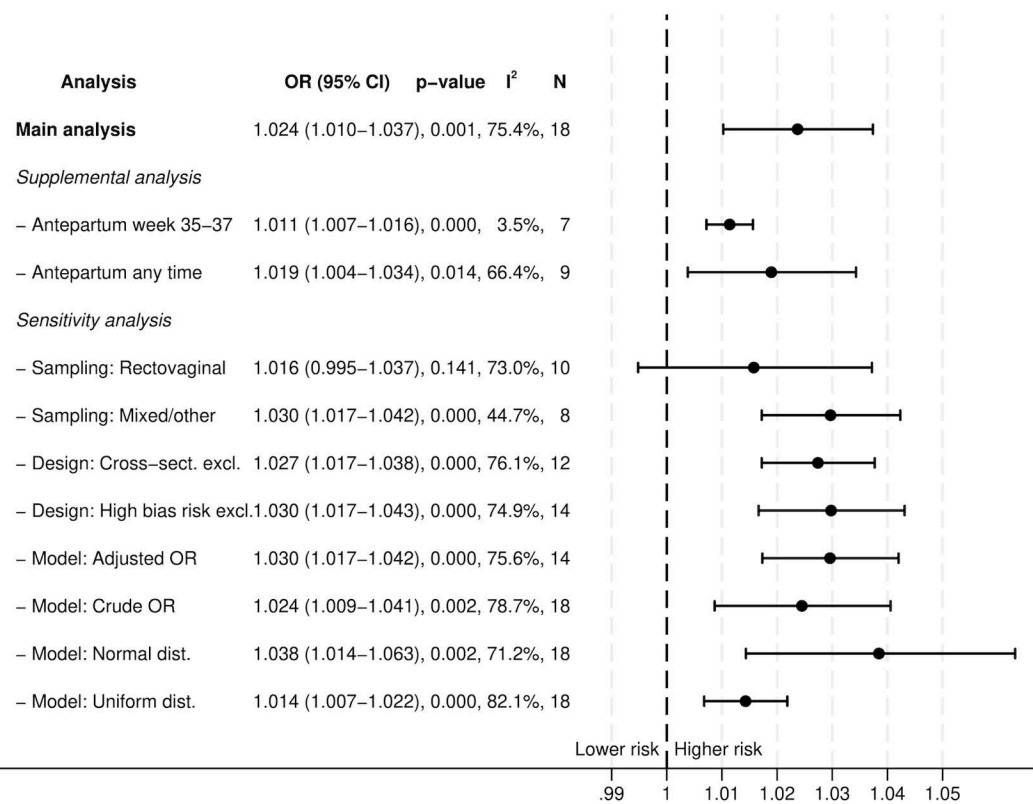

**Fig 3. Forest plot.** Forest plot representing the association between BMI and risk of EOGBS or proxy outcomes by odds ratios with 95% confidence intervals based on random-effects meta-regression on 18 studies represented with best linear unbiased predictors (BLUP) of the random-effects. Along with results from various supplementary and sensitivity analyses.

**Table 4. OR for outcome across selected BMI levels compared to normal level (BMI = 22.3) based on the main analysis including 18 studies and for the sensitivity analysis based on the 14 studies reporting adjusted ORs.**

|  | Main analysis | Adjusted |
|---|---|---|
| OR of GBS per Δ BMI | 1.024 (95% CI: 1.010–1.037) | 1.030 (95% CI: 1.017–1.042) |
|  | OR | Adjusted OR |
| BMI 22.3 (reference) | 1.0 | 1.0 |
| BMI 30 | 1.20 (95% CI 1.08–1.33) | 1.25 (95% CI 1.14–1.38) |
| BMI 35 | 1.35 (95% CI 1.14–1.60) | 1.45 (95% CI 1.25–1.69) |
| BMI 40 | 1.52 (95% CI 1.20–1.92) | 1.68 (95% CI 1.36–2.08) |
| BMI 45 | 1.71 (95% CI 1.26–2.30) | 1.94 (95% CI 1.48–2.55) |

**Table 5. QUIPS study quality assessment.**

| Study | Study participation | Study attrition | Prognostic factor (BMI) measurement | Outcome measurement | Study confounding | Statistical analysis and reporting | Bias risk in summary | Comments |
|---|---|---|---|---|---|---|---|---|
| Villamor, 2021 | ⊕ | ⊕ | ⊕ | ⊕ | ⊕ | ⊕ | Low | |
| Håkkanson, 2008* | ⊕ | ⊕ | ⊕ | ⊕ | ⊕ | ⊕ | Low | |
| Khalil, 2022 | ⊖ | ⊖ | ⊖ | ⊕ | ⊕ | ⊕ | Moderate | Baseline characteristics are sparse and response rate is only 46.6%. Lack of detailed information on exclusion reasons of 350 excluded participants. BMI assessment method not specified. These limitations are probably caused by the fact that this is a secondary analysis of a cohort. |
| Alvareza, 2017 | ⊖ | ⊖ | ⊖ | ⊖ | ⊕ | ⊕ | High | Risk of selection bias due to exclusion of cervix screening commitment and cervical excision procedures which can vary alongside socioeconomic status. No data on study attrition, but clear information on inclusion criteria. Lack of time and method of BMI assessment. No information on distributions between urinary and rectovaginal cultures or microbiological analyzing methods. |
| Rao, 2019 | ⊕ | ⊕ | ⊕ | ⊖ | ⊖ | ⊕ | Moderate | 6309 screened, but only 6131 included in multivariate analysis. No information on reasons for this loss of participants. Self-collected samples. |

*(Continued)*

| Study | Study participation | Study attrition | Prognostic factor (BMI) measurement | Outcome measurement | Study confounding | Statistical analysis and reporting | Bias risk in summary | Comments |
|---|---|---|---|---|---|---|---|---|
| Rick, 2017 | ⊖ | ⊕ | ⊖ | ⊕ | ⊕ | ⊕ | Moderate | Retrospectively collected data on children's outcomes in 73/155 cases. Risk of recall bias in BMI assessment, since the women were asked from GA 35. No information whether BMI was pregestational. |
| Namugongo, 2016 | ⊕ | ⊖ | ⊕ | ⊖ | ⊕ | ⊕ | Moderate | No data on response rate. Rapid strep B test kit only. No culture growth or PCR. |
| Bognar, 2025 | ⊕ | ⊕ | ⊖ | ⊖ | ⊕ | ⊕ | Moderate | BMI "before pregnancy" from the Belgium Birth Register (BBR). GBS colonization positive outcome retrieved from the BBR is not specified. |
| Zhou, 2024 | ⊕ | ⊖ | ⊖ | ⊕ | ⊖⊖ | ⊖ | High | Risk of selection bias cannot be ruled out, inclusion criteria reduces generalizability. Lack of information on BMI assessment. No adjustment for confounders. |
| Nguyen, 2025 | ⊕ | ⊕ | ⊖ | ⊕ | ⊖ | ⊖ | Moderate | Multivariate logistic regression analyses were conducted. |
| Manzanares, 2012 | ⊕ | ⊖ | ⊕ | ⊕ | ⊕ | ⊕ | Low | Selection bias might occur since only 3016 out of 12.781 (24%) singleton pregnancies had available pregestational BMI data. |
| Manzanares, 2019 | ⊕ | ⊕ | ⊕ | ⊕ | ⊕ | ⊕ | Low | |
| Kwon, 2023 | ⊕ | ⊕ | ⊖ | ⊕ | ⊖⊖ | ⊕ | High | Limited generalizability due to black population. Self-reported BMI, 20% missing BMI data. Missing information on pregestational BMI. Risk of study confounding from Black population and crude estimates, though this risk was partly diminished by excluding women with diabetes, GDM and chronic hypertension. |
| Dahan-Saal, 2011 | ⊕ | ⊖ | ⊖ | ⊕ | ⊕ | ⊕ | Moderate | Well-explained participant selection process, but >50% of eligible GBS tested women excluded due to lack of data. No information on method/time of BMI assessment. |
| Kleweis, 2015 | ⊕ | ⊕ | ⊖⊖ | ⊕ | ⊖ | ⊕ | High | BMI assessed at admission with no clear information on whether BMI was pregestational. If not, this could be an explanation for the large proportion of severely obese women in this cohort. We wrote to the authors for clarification but had no response. |

*(Continued)*

**Table 5.** (Continued)

| Study | Study participation | Study attrition | Prognostic factor (BMI) measurement | Outcome measurement | Study confounding | Statistical analysis and reporting | Bias risk in summary | Comments |
|---|---|---|---|---|---|---|---|---|
| Venkatesh, 2020 | ⊕ | ⊕ | ⊕ | ⊖ | ⊕ | ⊕ | Low | Timing and specification of sample site not available. |
| Stapleton, 2005 | ⊕ | ⊖ | ⊕ | ⊕ | ⊕ | ⊕ | Low | Prevalence only 10.8%, possibly due to misclassification bias since identification of cases were based on ICD-9 codes, and "not screened" could not be differentiated from "not colonized". |
| Chen, 2018 | ⊕ | ⊕ | ⊕ | ⊖ | ⊕ | ⊕ | Low | Only vaginal swab used which might lead to an underestimation of the true prevalence. |
| Melchor, 2019 | ⊕ | ⊖ | ⊕ | ⊕ | ⊖ | ⊕ | Moderate | Missing information on study attrition. |

*This study was not included in the primary analysis due to overlapping cohorts with Villamor et al, 2021 [7].

⊕Low risk of bias in the studied domain, ⊖ Moderate risk of bias in the studied domain, ⊖⊖ High risk of bias in the studied domain.

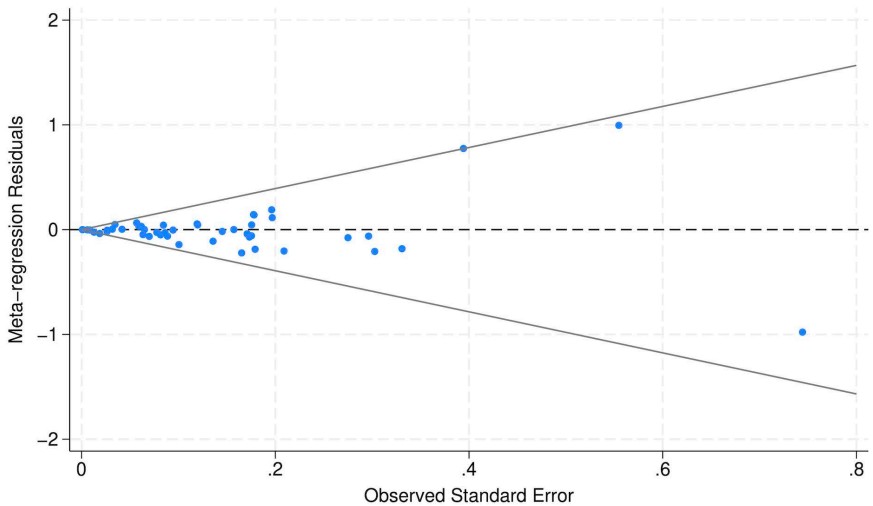

**Fig 4. Funnel plot.** Regression funnel plot including all studies (N = 18), representing 63 data points in the primary random-effects meta-regression analysis. Residuals, expressed on the log odds ratio scale, from the fitted model are plotted against their corresponding observed standard errors. The vertical line indicates zero residual (no deviation from the fitted regression line), and the dashed lines represent the approximate 95% confidence limits. Asymmetry in the dispersion of residuals across levels of precision may indicate small-study effects; however, such patterns may also reflect between-study heterogeneity or model misspecification.

## 4. Discussion

### 4.1 Principal findings

This systematic review suggests a dose–response relationship between increasing preconception BMI and a higher risk of EOGBS, intrapartum vGBS, and rectovaginal GBS colonization. Using a BMI of 22.3 as the reference, the association for this composite outcome corresponds to an OR of 1.20 at BMI 30, 1.35 at BMI 35, 1.52 at BMI 40, and

1.71 at BMI 45. For the most clinically important outcome, EOGBS, which was addressed in two overlapping studies, the adjusted hazard ratio in the largest cohort was 2.6 for women with a BMI ≥ 40 [7]. Although absolute risks are low, the estimated association with BMI may represent a clinically relevant difference in risk for higher levels of BMI. In the example described above, a BMI level of 40 represents an excess risk of 0.22 (95% CI 0.07–0.37) per 1000 (S2 Table), although due to the uncertainty as represented in this meta-analysis, this may be as low as 0.07 per 1000 above the level of the normal group.

The biological mechanisms linking increased BMI and EOGBS proxy outcomes are not yet fully understood, but several hypotheses exist. The association between obesity and EOGBS may involve specific pathways, including the interaction of dietary fats, such as palmitate, which can induce inflammatory responses in the gestational membranes and the placenta [35] potentially increasing susceptibility to GBS colonization [36]. Obesity alters immune responses, gut and vaginal microbiota, and inflammatory pathways, all of which may predispose individuals to persistent GBS colonization [37]. Additionally, metabolic dysregulation common in obesity may impair mucosal defences, thereby facilitating GBS persistence and vertical transmission to the infant during labor.

Although BMI is not a standard risk factor in many existing clinical guidelines for EOGBS, both research and clinical practice support our findings that high maternal BMI may increase the risk of EOGBS [38–40]. Besides obesity, GDM and pregestational diabetes are novel risk factors that have been studied. A recent meta-analysis from 2024 found GDM was associated with a 16% elevated risk of maternal GBS colonization, and an even larger risk of 76% reported for pregestational diabetes (OR 1.76, 95% CI 1.27–2.45) [41].

## 4.2 Clinical implications

When considering the rationale for including BMI as a risk factor in EOGBS prevention programs, it is essential to consider all aspects of a medical technology evaluation. In addition to clinical effectiveness, these include safety (e.g., allergic reactions and microbiome alterations), risks (e.g., bacterial resistance to antibiotics), costs (both short- and long-term), implementability (e.g., training requirements for healthcare staff), patient perspectives, ethical implications (e.g., equity), and alternative approaches (e.g., vaccination or intrapartum assessments of vGBS status by PCR techniques).

Further, one must acknowledge that most of the traditional risk factors in the risk-based screening model are notably stronger than the BMI-related risk found in this review; e.g., prolonged rupture of membranes > 18 hours (OR: 7.3), intrapartum fever (OR 4.1) or PROM at term (OR 11.5 with PROM or intrapartum fever – or both – present) and gestation < 37 weeks (OR: 4.8) [1]. However, including BMI in risk stratification may reduce the number of missed cases. In Denmark, for instance, women with a BMI > 35 account for 5.7% of the birthing population [42]. These women may face a 10% risk of a false-negative risk stratification [43,44].

## 4.3 Strengths and limitations

This study adheres to rigorous systematic review methodology, including comprehensive search, bias assessment, duplicate screening, structured RoB assessment, and reporting in accordance PRISMA standards. A major strength is the inclusion of diverse populations from high-, middle-, and low-income countries, enhancing external validity and generalizability. Furthermore, including large cohorts across diverse study designs increased the overall sample size and statistical power. However, substantial methodological and clinical heterogeneity must be acknowledged. The included studies differed in design (cohort, cross-sectional, and case-control), population characteristics, GBS screening strategies, BMI ascertainment, and statistical adjustment models. Although the use of a random-effects model accounts for between-study variance statistically, it does not eliminate underlying clinical or methodological heterogeneity. Consequently, pooled estimates should be interpreted as average effects across heterogeneous contexts rather than precise effect sizes applicable to all settings. Design-related heterogeneity is particularly important. Cross-sectional studies do not establish temporality,

however subgroup analyses showed no weakening of the association when cross-sectional studies were excluded: OR 1.027 (95% CI 1.017–1.038, p value < 0.001).

Furthermore, case-control studies may be prone to selection bias. Although cohort studies provide stronger temporal inference, the observational nature of all included studies limits causal interpretation. We therefore emphasize that the findings demonstrate association rather than causation.

Risk of bias was primarily driven by potential confounding and measurement variability. Although many studies adjusted for key confounders such as maternal age, parity, diabetes, and smoking, adjustment strategies varied considerably, and several studies did not account for racial composition or behavioral and socioeconomic factors. The present study was not powered to assess potential effect modification by ethnicity [45,46], differences in health care systems, or socioeconomic status [26]. Notably, a stronger association among Black women has been consistently reported across several studies [10,23,45–48]. Stratification by adjustment status (adjusted vs. unadjusted estimates) did not materially alter the pooled results, suggesting that confounding is unlikely to fully explain the observed association. However, given the limited adjustment strategies in several included studies, residual confounding cannot be ruled out (Fig 3 and Table 3).

Measurement heterogeneity further contributes to the risk of bias. GBS ascertainment varied across studies in terms of testing protocols, including differences in culture versus PCR testing, swab collection techniques, and gestational timing. Similarly, the timing of BMI measurement varied across the included studies, introducing potential misclassification bias. Moreover, we assume that BMI assessment was commonly based on self-reporting, which may introduce recall bias. In many registry-based and historical cohort studies, the timing of BMI assessment was insufficiently specified, precluding subgroup analyses and limiting exploration of this source of heterogeneity. Limitations in the statistical strategy used may include the approach to variation between observations, which involved translating reported BMI categories into continuous-scale intervals based on data from the Danish Medical Birth Registry. However, this allowed for a rigorous meta-analysis with results spanning a wider range of BMI values, thereby enabling the proof of dose-response associations, a strength of this review.

In summary, this review is strengthened by methodological rigor, a large aggregated sample, and dose-response modeling. However, substantial clinical and methodological heterogeneity, variability in adjustment strategies, and inherent limitations of observational data introduce risk of bias and restrict causal interpretation. These findings should therefore be interpreted as robust associative evidence, warranting cautious inference and further high-quality prospective research.

### 4.4 Future directions

Additional research is needed to clarify the biological mechanisms linking obesity and GBS colonization and to refine screening guidelines to optimize the balance between effectiveness and safety [49]. Topics of interest could be the roles of the microbiome, inflammation, and metabolic pathways. Additionally, explore patient perspectives and ethical implications of revised prevention protocols. Moreover, trials are needed to assess whether lifestyle interventions, such as weight optimization or targeted probiotics, reduce GBS colonization and transmission. Lastly, modelling studies may help evaluate the cost-effectiveness and safety of incorporating BMI into GBS screening guidelines.

### 5. Conclusion

This systematic review and meta-analysis suggests a modest but consistent association between increasing maternal BMI and the risk of GBS colonization and early-onset GBS disease. Therefore, the integration of BMI into existing risk assessment models warrants consideration, pending further evidence and implementation analysis.

### Supporting information

**S1 Table. Search strategy.**
(PDF)

**S2 Table. Absolute risk by BMI level.**
(DOCX)

**S1 File. PRISMA Checklist (2020).**
(DOCX)

**S2 File. Data set (CSV).** Supporting information on study identification, BMI categories, effect measures, and 95% CIs used in the main analysis.
(CSV)

**S1 Fig. Leave-one-out analyses.**
(DOCX)

## Acknowledgments

We thank Jeppe Benneskou Schroll (Center for Evidence-Based Medicine Odense) and Lasse Østengaard (University of Southern Denmark) for their advice on the conceptualization of this systematic review. We also thank Torben Bjerregaard Larsen for guidance on planning the statistical analysis, and Peter Everfelt (Esbjerg and Grindsted Hospital) for initial support with the literature search. The contributors acknowledged here did not participate in the writing of the manuscript or in the decision to submit it for publication, and may not necessarily agree with its content.

## Author contributions

**Conceptualization:** Elisabeth Emanuel Graae, Niels Uldbjerg, Flemming Skjøth, Caroline Margaret Moos, Mohammed Rohi Khalil.

**Data curation:** Elisabeth Emanuel Graae, Flemming Skjøth.

**Formal analysis:** Niels Uldbjerg, Flemming Skjøth.

**Investigation:** Elisabeth Emanuel Graae, Marie Aunskjær Bech, Pernille Nathalie Nielsen, Susani Rothmann Karkov, Caroline Margaret Moos, Mohammed Rohi Khalil.

**Methodology:** Elisabeth Emanuel Graae, Flemming Skjøth, Caroline Margaret Moos.

**Project administration:** Elisabeth Emanuel Graae, Niels Uldbjerg, Mohammed Rohi Khalil.

**Software:** Flemming Skjøth.

**Supervision:** Niels Uldbjerg, Flemming Skjøth, Caroline Margaret Moos, Mohammed Rohi Khalil.

**Validation:** Elisabeth Emanuel Graae, Niels Uldbjerg.

**Visualization:** Elisabeth Emanuel Graae.

**Writing – original draft:** Elisabeth Emanuel Graae, Mohammed Rohi Khalil.

**Writing – review & editing:** Elisabeth Emanuel Graae, Niels Uldbjerg, Flemming Skjøth, Mohammed Rohi Khalil.

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
