## [Editor Report · Decision Letter 0]

27 Oct 2025

PONE-D-25-38579Maternal Body Mass Index and the Risk of Early-Onset Group B Streptococcus Disease in Newborns: A Systematic Review and Meta-AnalysisPLOS ONE

Dear Dr. Graae,

Thank you for submitting your manuscript to PLOS ONE. After careful consideration, we feel that it has merit but does not fully meet PLOS ONE’s publication criteria as it currently stands. Therefore, we invite you to submit a revised version of the manuscript that addresses the points raised during the review process.

We look forward to receiving your revised manuscript.

Kind regards,

Ho Yeon Kim

Academic Editor

PLOS ONE

Journal Requirements:

2. We note that your Data Availability Statement is currently as follows: [All relevant data are within the manuscript and its Supporting Information files]Please confirm at this time whether or not your submission contains all raw data required to replicate the results of your study. Authors must share the “minimal data set” for their submission. PLOS defines the minimal data set to consist of the data required to replicate all study findings reported in the article, as well as related metadata and methods (https://journals.plos.org/plosone/s/data-availability#loc-minimal-data-set-definition).

Additional Editor Comments (if provided):

This is a valuable and timely systematic review and meta-analysis, providing significant evidence of an association between maternal BMI and GBS-related outcomes. The findings have important implications for improving prediction and prevention strategies in obstetric care. However there are some major issues in methodology. If these issues are not addressed, it will be difficult to accept the manuscript.

1. BMI Categorization and Transformation : Clarify how sensitive the results are to this assumption, or consider performing sensitivity analysis where different distributions are assumed.

2. Inclusion of Various Study Designs : The review includes cohort, cross-sectional, and case-control studies. These designs have different strengths and limitations, especially regarding causal inference.

Combining these designs in meta-regression may introduce heterogeneity. While random-effects models help account for this, explicit acknowledgment and discussion on how this diversity impacts pooled estimates would be helpful.

3. Outcome Definitions and Testing Methods

The methods mention variability in GBS testing protocols (culture vs PCR, timing during pregnancy, anatomical sampling sites). Explicitly describe how this variability was addressed or explored via subgroup analyses, which they partially do.

4. Adjustments and Confounders

Heterogeneity in adjustment strategies may influence the comparability and validity of pooled estimates. Specify the key confounders considered and whether there was a pre-specified approach to select the adjusted ORs for meta-analysis.

5. Statistical Methodology : Clearly state whether model diagnostics were performed to assess fit and assumptions.

---

## [Author Response · Author response to Decision Letter 1]

21 Nov 2025

Dear Editor,

Please find enclosed our revised manuscript entitled “Maternal Body Mass Index and the Risk of Early-Onset Group B Streptococcus Disease in Newborns: A Systematic Review and Meta-Analysis.” We have carefully addressed all comments, as detailed in the attached Rebuttal Letter. The revised version includes expanded methodological descriptions, sensitivity analyses, and updated data-sharing information in full compliance with PLOS ONE requirements.

We sincerely thank you for considering our revision and look forward to your evaluation.

Best regards,

Elisabeth Emanuel Graae, MD

Corresponding Author

---

## [Decision Letter · Decision Letter 1]

21 Jan 2026

PONE-D-25-38579R1Maternal Body Mass Index and the Risk of Early-Onset Group B Streptococcus Disease in Newborns: A Systematic Review and Meta-AnalysisPLOS One

Dear Dr. Graae,

Thank you for submitting your manuscript to PLOS ONE. After careful consideration, we feel that it has merit but does not fully meet PLOS ONE’s publication criteria as it currently stands. Therefore, we invite you to submit a revised version of the manuscript that addresses the points raised during the review process.

We look forward to receiving your revised manuscript.

Kind regards,

Ho Yeon Kim

Academic Editor

PLOS One

**Journal Requirements:**

Reviewers' comments:

Reviewer's Responses to Questions

**Comments to the Author**

1. If the authors have adequately addressed your comments raised in a previous round of review and you feel that this manuscript is now acceptable for publication, you may indicate that here to bypass the “Comments to the Author” section, enter your conflict of interest statement in the “Confidential to Editor” section, and submit your "Accept" recommendation.

Reviewer #1: (No Response)

Reviewer #2: (No Response)

Reviewer #3: (No Response)

2. Is the manuscript technically sound, and do the data support the conclusions?

Reviewer #1: Partly

Reviewer #2: Yes

Reviewer #3: (No Response)

3. Has the statistical analysis been performed appropriately and rigorously? 

Reviewer #1: No

Reviewer #2: Yes

Reviewer #3: (No Response)

4. Have the authors made all data underlying the findings in their manuscript fully available?

Reviewer #1: No

Reviewer #2: Yes

Reviewer #3: Yes

5. Is the manuscript presented in an intelligible fashion and written in standard English?

Reviewer #1: No

Reviewer #2: Yes

Reviewer #3: Yes

6. Review Comments to the Author

**Reviewer #1:** The subject addressed in this systematic review is of considerable importance. Unfortunately, there are major weaknesses in the report. The authors may find the Handbook produced by the Cochrane Collaboration (cochrane.org/authors/handbooks-and-manuals/handbook/current) helpful. My detailed comments are as follows:

MAJOR

1. p.3, ll.2-4. For the benefit of non-clinical readers, more background information should be given regarding Group B Streptococcus and, specifically, EOGBS. In addition, the distinction between parity and gravidity should be described.

2. p.4, ll.19-20. The time periods covered by the MEDLINE, Embase and CENTRAL databases should be stated. Given the date of the search (5th August 2024), it needs to be re-run.

3. p.6. References should be given for random-effects meta-regression (l.3) leave-one-out analysis (l.19) and the I-squared statistic (bottom line).

4. pp.9-10, Table 1. The reference numbers given in the first column do not match the corresponding numbers in Tables 2 and 3, e.g. Rao (2019) is [13] in Table 1 but [14] in Tables 2 and 3. In Column 4, "N (cases)" might be clearer as "Sample size (cases)".

5. pp.11-13, Table 2. All papers cited in this table should be carefully checked for content as important information is missing, e.g., in Rao (2019) the percentages for the ethnic groups considered are reported (White British 27.4%, White other 32.0%, etc). NB I have not checked all the papers. At the top of Column 6, what does "P0 %" mean? The information on socio-economic status (Column 8) is confusing, as education, marital status, an index of deprivation and type of insurance are all in this single column. The demographic information should be presented in an additional table, with separate columns for education, marital status, etc. For Rao (2019), the acronym IMD should be explained.

6. pp.13-15, Table 3. The information presented in this table is very dense, and the Main Results (Column 4) are particularly difficult to identify. aHR, aOR and OR should be explained. There should be separate columns for BMI category, the estimate, and 95% confidence interval. This might be easier to achieve with the table in landscape format.

7. p.19, ll.20-21. I find this statement regarding the preference for a crude OR concerning, given that there is potentially a high risk of seriously misleading findings if the effects of other appropriate explanatory variables are ignored.

8. p.21, ll.4-6. A clear distinction should be drawn between statistical significance and clinical importance. In determining the clinical importance of a finding, the limit of the 95% confidence interval closer to the null hypothesis value, in this case the lower limit, should be considered.

9. In Figure 1 (PRISMA flow-chart), the numbers of records identified, removed because of duplication and screened are incompatible, i.e., 844 + 491 + 311 + 42 - 624 is not equal to 689. Is there a typing error here?

MINOR

1. p.5, l.12. The role of the individuals involved in the third-party adjudication should be stated, e.g., were they part of the project team? Did they input into other aspects of this work?

2. p.5, l.12. A reference should be given for the Covidence software.

3. p.5, ll.16-17. The primary outcome and each proxy outcome should be listed here using bullet points.

4. p.5. Were the corresponding authors of the relevant reports contacted in order to obtain additional information?

5. p.29. Should "syntesis" be "synthesis"?

6. The Figures are not presented in the correct order.

7. In Figure 4 (QUIPS study quality assessment) what do the green and light orange circles represent?

**Reviewer #2:** The manuscript addresses a pertinent clinical question and is generally well organized. However, I have four comments regarding reporting that could enhance clarity and methodological transparency.

1. Citation Numbering in Table 1

In Table 1, the numeric citation labels for several studies do not match the reference list numbering. For instance, Namugongo 2016 is labeled as “(15)” in Table 1, while it is referenced as “(16)” in the reference list. In contrast, the citations in the other tables are accurately numbered, which suggests a formatting issue specific to Table 1. To prevent misunderstandings for readers who wish to verify the studies included, please correct Table 1 to ensure that the citation numbering aligns with the reference list.

2. PRISMA Flow Diagram Arithmetic

The PRISMA flow diagram shows an inconsistency in the numbers reported for identified records, duplicates removed, and records screened. Specifically, the diagram indicates Records identified (n = 844), Duplicates removed (n = 624), and Records screened (n = 689); however, (844 − 624 does not equal 689) . This step needs to be recalculated, and the PRISMA diagram should be updated accordingly.

3. Clarification of BMI Measurement Timing

It would be beneficial to clarify whether BMI was assessed prior to pregnancy, at first antenatal booking, or at later stages of gestation for each of the included studies. These time points are not interchangeable clinically, and pooling these measures without clear delineation may lead to exposure misclassification. Even if subgroup analysis is not feasible, specifying the timing in the Methods or Tables and briefly discussing this limitation in the Discussion section would enhance interpretability.

4. Funnel Plot Description and Interpretation

In describing Fig. 5, it states that it represents a funnel plot of individual study log odds ratios against their standard errors to assess publication bias. However, the axes appear to depict residuals versus standard error, which aligns more closely with influence diagnostics than with a traditional funnel plot. Additionally, the figure label claims the plot is based on “all included studies,” yet it remains unclear which outcome(s) these studies pertain to. For the primary outcome (EOGBS), the small number of included studies (approximately k = 5) typically does not provide a robust basis for funnel plot analysis. Clarifying the dataset represented and tempering the interpretation would significantly enhance the reporting.

**Reviewer #3:** Methods

• Section 2.3: the statement that no studies were yielded from the grey literature/citation tracking should not be found in the methods. I suggest omitting or moving to the results. There is an extra period after “AND.” Inconsistent capitalization of medical subject headings. Suggest defining MeSH as it is first introduced without defining the acronym. Additional information on grey literature search would be appreciated (e.g., how many studies searched, what was searched).

• Section 2.4: The entire section on PFO criteria should be moved to the eligibility criteria. The criteria should be presented in a paragraph, not in bulleted text. Clarify who was responsible for “third-party adjudication” and if data extraction was performed in duplicate (if so, if there was a process for resolving discrepancies for data extraction).

• Section 2.5: citation for the source of the numbers log 24.6 and log 1.2. OR needs to be explicitly defined. If there were prespecified thresholds for heterogeneity, they should be specified. What does “No major violations were identified” refer to? It’s not clear and also sounds like a result, not a prespecified method. Specify packages used in Stata for the analysis. Unclear what the purpose of a sensitivity analysis of OR < 2 is for.

• Strongly recommend adopting the GRADE framework for assessing certainty in the evidence for improved interpretation of the findings. It would also be beneficial to have absolute risk differences to help contextualize the absolute magnitude of some of the relative effect estimates you find.

Results

• Section 3.1: Avoid using ‘e.g.’ in a sentence. For the categorization of GBS assessment timing, specify the number and % of studies for each. In the sentence describing geographical region, use textual numbers for all for consistency and because the values < 10. P values should be reported throughout. I2 should be provided for major results in the text. What does “exponential increase” in OR mean? The main finding (OR 1.027) needs to be clearly reported in the text with 95% CI. All subgroup analyses should report a p value for interaction to quantify whether there are significant differences between subgroups.

• Section 3.2: details about RoB assessment should be first introduced in the methods. A citation is needed for the QUIPS tool. Specify the authors who conducted the assessment and what process was used to resolve conflicts. Define what your overall risk of bias levels are for the studies and how these were determined from the individual domains. Specify the number and % of the low RoB studies when you refer to “most studies.” I have serious doubts about the quality of the QUIPS assessment. All studies are rated to be at a low risk of bias due to confounding, but many studies report unadjusted estimates which are very vulnerable to confounding.

• Figure 1: the numbers in records identified, duplicates removed, and records screened to not add up.

• Figure 3: suggest avoiding arrows at the end of 95% CIs unless the CI extends beyond the axis labels. Also suggest having a separate forest plot for the pooled analysis that shows the ORs of the individual studies along with the pooled estimate. This new forest plot should show the weights of the included studies.

Discussion

• Section 4.3: further discussion on heterogeneity is warranted. For example, the study characteristics show notable difference in racial composition, and GBS colonization rates and BMI may vary across these populations.

• An exploration of physiological reasons why EOGBS may be increased in patients who are obese would be relevant.

Supplemental information

• Provide the number of results per line in all search strategies

• Ensure that Table S5 is CSV, it appears to be a Numbers file

Formatting

• Line numbers, which are required by PLOS One, would be very helpful for review comments

• Suggest adding a space between < or > and the following number throughout

7. PLOS authors have the option to publish the peer review history of their article (what does this mean?). If published, this will include your full peer review and any attached files.

Reviewer #1: No

Reviewer #2: **Yes:** Alousious Kasagga

Reviewer #3: **Yes:** David Gou

---

## [Author Response · Author response to Decision Letter 2]

13 Mar 2026

Please see the attached file "Response to Reviewers" in which all comments are carefully addressed.

---

## [Decision Letter · Decision Letter 2]

13 Apr 2026

PONE-D-25-38579R2Maternal Body Mass Index and the Risk of Early-Onset Group B Streptococcus Disease in Newborns: A Systematic Review and Meta-AnalysisPLOS One

Dear Dr. Graae,

Thank you for submitting your manuscript to PLOS ONE. After careful consideration, we feel that it has merit but does not fully meet PLOS ONE’s publication criteria as it currently stands. Therefore, we invite you to submit a revised version of the manuscript that addresses the points raised during the review process.

We look forward to receiving your revised manuscript.

Kind regards,

Ho Yeon Kim

Academic Editor

PLOS One

**Journal Requirements:**

Reviewers' comments:

Reviewer's Responses to Questions

**Comments to the Author**

1. If the authors have adequately addressed your comments raised in a previous round of review and you feel that this manuscript is now acceptable for publication, you may indicate that here to bypass the “Comments to the Author” section, enter your conflict of interest statement in the “Confidential to Editor” section, and submit your "Accept" recommendation.

Reviewer #1: (No Response)

Reviewer #3: (No Response)

2. Is the manuscript technically sound, and do the data support the conclusions?

Reviewer #1: Yes

Reviewer #3: Yes

3. Has the statistical analysis been performed appropriately and rigorously? 

Reviewer #1: Yes

Reviewer #3: Yes

4. Have the authors made all data underlying the findings in their manuscript fully available?

Reviewer #1: Yes

Reviewer #3: Yes

5. Is the manuscript presented in an intelligible fashion and written in standard English?

Reviewer #1: Yes

Reviewer #3: Yes

6. Review Comments to the Author

Reviewer #1: The manuscript is much better. I have a few minor comments:

1. ll. 199-203. This sentence is difficult to follow. Are the semicolons and commas in the correct positions? A semicolon might be more appropriate after "[22]" and [12,23-28]".

2. Table 2. There are two instances of "marietal status".

3. l. 403. The word "diversity" could be interpreted in the context of EDI (Equality, Diversity, and Inclusion). Might "variation between observations" be better?

4. ll. 426-427. Is this sentence complete?

Reviewer #3: Most of the previous comments appear to be addressed. I still find the pre-planned sensitivity analysis restricting studies based on OR < 2 to be unjustified. Excluding studies solely based on the magnitude of their effect estimates introduces bias; if studies with more extreme effect estimates were thought to be at a higher risk of bias, this would be accounted for in the analysis by study risk of bias. Leave-one-out analyses are already performed to assess the stability of the pooled results. Additionally, there is no justification why a threshold of OR = 2.0 was selected. Your protocol does not state this threshold was established a priori: “Additionally, supplementary sensitivity analyses will be conducted, to evaluate the influence of factors such as large studies on the overall effect measure in cases of heterogeneity.” I would argue that none of the sensitivity analyses that were performed were explicitly pre-planned, and describing them as such may be misleading.

Formatting: line 158 I^2 should be in superscript.

7. PLOS authors have the option to publish the peer review history of their article (what does this mean?). If published, this will include your full peer review and any attached files.

Reviewer #1: No

Reviewer #3: **Yes:** David Gou

---

## [Author Response · Author response to Decision Letter 3]

16 Apr 2026

Our detailed response letter addressing each comment is attached for your review.

---

## [Editor Report · Decision Letter 3]

19 Apr 2026

Maternal Body Mass Index and the Risk of Early-Onset Group B Streptococcus Disease in Newborns: A Systematic Review and Meta-Analysis

PONE-D-25-38579R3

Dear Dr. Graae,

We’re pleased to inform you that your manuscript has been judged scientifically suitable for publication and will be formally accepted for publication once it meets all outstanding technical requirements.

Kind regards,

Ho Yeon Kim

Academic Editor

PLOS One

Additional Editor Comments (optional):

Reviewers' comments: The revisions have been addressed appropriately.

---

## [Editor Report · Acceptance letter]

PONE-D-25-38579R3

PLOS One

Dear Dr. Graae,

I'm pleased to inform you that your manuscript has been deemed suitable for publication in PLOS One. Congratulations! Your manuscript is now being handed over to our production team.

Kind regards,

on behalf of

Professor Ho Yeon Kim

Academic Editor

PLOS One